# Evaluation of Neoplasia, Treatments, and Survival in Lizard Species

**DOI:** 10.3390/ani14101395

**Published:** 2024-05-07

**Authors:** Frank Willig, Fred J. Torpy, Scott H. Harrison, Elizabeth G. Duke, Brigid Troan, Amy M. Boddy, Lisa M. Abegglen, Tara M. Harrison

**Affiliations:** 1Virginia Maryland College of Veterinary Medicine, Blacksburg, VA 24061, USA; 2Exotic Species Cancer Research Alliance, North Carolina State University College of Veterinary Medicine, Raleigh, NC 27607, USAecgraebe@ncsu.edu (E.G.D.); bvtroan@ncsu.edu (B.T.); amyboddy@ucsb.edu (A.M.B.); lisa.abegglen@hci.utah.edu (L.M.A.); 3Department of Small Animal Medicine and Surgery, University of Georgia College of Veterinary Medicine, Athens, GA 30602, USA; 4Department of Biology, North Carolina Agriculture and Technical State University, Greensboro, NC 27411, USA; scotth@ncat.edu; 5Department of Clinical Sciences, North Carolina State University College of Veterinary Medicine, Raleigh, NC 27607, USA; 6Department of Molecular and Biomedical Sciences, North Carolina State University College of Veterinary Medicine, Raleigh, NC 27607, USA; 7Department of Anthropology, University of California Santa Barbara, Santa Barbara, CA 93106, USA; 8Department of Pediatrics, University of Utah, Salt Lake City, UT 84112, USA

**Keywords:** neoplasia, lizard, surgery, chemotherapy, bearded dragon, iguana, monitor, lymphoma, carcinoma

## Abstract

**Simple Summary:**

Neoplasia is a complex disease that affects many species across the animal kingdom, including lizards. Currently, cancer in lizard species is an understudied part of veterinary medicine. In this study, we focused on identifying factors that could aid in improving patient care and quality of life for lizards with neoplasia. We identified multiple factors including species, type of neoplasia, and type of treatment significantly associated with both positive and negative outcomes for lizards affected by different types of neoplasia. Specifically, we tested for statistical associations between eight clinical factors and patient outcomes. We used reported cases of neoplasia in lizards from published papers, as well as a clinical oncology database for exotic animal species. We also identified a subset of neoplasia types that were not associated with death due to their neoplasia. Our results highlight the importance of determining variables that aid veterinarians in deciding the most appropriate care for their patients. We expect that future research in this area will improve our understanding of neoplasia in lizards and better improve the identification of predictor variables for improving patient outcomes.

**Abstract:**

Neoplasia has been reported in lizards, but more research is needed to accurately document the prevalence and prognosis of the various known neoplasms that affect lizards. This study reviewed medical records from an online database, the Exotic Species Cancer Research Alliance (ESCRA), and reviewed published literature to determine the prevalence of neoplasia, malignancy, metastasis, treatment strategies, and outcomes by species and sex. Records from 55 individual lizards, 20 different species, and 37 different tumors were identified. In the literature, 219 lizards, 59 species, and 86 unique tumors were identified from 72 published case reports. Potential signalment factors such as age, sex, and species were evaluated to see if they affected case outcome. Additional factors including neoplasia type, presence of metastasis, and types of pursued treatments were also evaluated. Statistical analysis was performed to determine whether a factor was significantly associated with animal death due to the identified neoplasia or with animal survival or death due to other causes (non-neoplastic outcomes). Komodo dragons and savannah monitors were more likely to die from neoplasia compared to other lizard species. Cases where the status of metastasis was unknown were significantly associated with death due to neoplasia. Having an unknown status of male versus female was significantly associated with non-neoplastic outcomes of death. Leukemia and islet cell carcinoma were significantly associated with death due to neoplastic causes. Chondrosarcoma, myxosarcoma, osteosarcoma, and squamous cell carcinoma were significantly associated with non-neoplastic outcomes of death. Surgery alone and radiation therapy alone each were significantly associated with non-neoplastic outcomes of death, while lizards not receiving treatment were significantly associated with death due to neoplasia. Benign neoplasia was significantly associated with non-neoplastic outcomes of death. These results will aid in the improved diagnosis and management of neoplasia in lizard species, as well as expanding our understanding of prognostic indicators of neoplasia in lizards.

## 1. Introduction

Neoplasia in lizards has been extensively documented going back well into the previous century consisting of mostly single case reports and a handful of reviews [1,2]. Neoplasia is common in lizards and a wide variety of neoplasms have been diagnosed. Similar to other taxa, various etiologies have been proposed, such as environmental, genetic, and infectious causes [3,4,5,6,7]. Several reports hypothesize species predispositions for certain neoplasms such as gastric neuroendocrine carcinoma in bearded dragons and pancreatic masses in Komodo dragons [4,8]. As advanced care is increasingly utilized in lizards, treatment is also more commonly being pursued for neoplasia. The therapeutic approaches for cancer reported in lizards include chemotherapeutics, radiation therapy, and surgical excision [9,10,11]. Though treatment modalities are described, comprehensive data on treatment efficacy and outcomes are lacking. Efforts to gather further information on neoplasia prevalence, treatment, and outcomes are ongoing. The Exotic Species Cancer Research Alliance (ESCRA) is a multi-institution database that collects clinical data on neoplasia in zoological species. The existence of such databases in other fields, such as the National Cancer Institute Data Catalogue, increases access to relevant information for researchers and promotes broader analyses. This database also promotes the standardization of information recording and reporting within the field.

This study combined case data from the ESCRA database with available published cases to determine the described prevalence of specific neoplasms, as well as to document the efficacy of treatment approaches by measuring outcomes. In addition, this summary of lizard neoplasia data from the literature can benefit researchers and clinicians alike.

## 2. Materials and Methods

### 2.1. Case Selection

#### 2.1.1. Literature Review

Databases including PubMed, CAB Abstracts, and Web of Science were searched for lizard neoplasia cases from the dates of database inception (1983 for PubMed, 1973 for CAB, and 1865 for Web of Science). Final searches for appropriate publications were performed between 27 July and 6 August 2021. Lizards that had experimentally induced tumors were excluded from this study. Search criteria were related to neoplasia in lizards, using terms for species within the order Squamata and including scientific names, common names, synonyms, and alternate spellings. The list of terms for neoplasia based on PubMed’s cancer subset search strategy was also utilized in this search. The described search criteria yielded manuscript results as follows: 28 manuscripts from PubMed met the inclusion criteria for this study, of which 20 were unique from CAB Abstracts and Web of Science; 51 manuscripts were identified from Web of Science, of which 13 were unique from the CAB Abstracts and PubMed manuscripts; and 73 manuscripts were identified from CAB Abstracts, of which 39 were unique from the PubMed and Web of Science manuscripts. This yielded a combined total of 72 unique manuscripts ranging from 1978 to 2021.

Data were imported from these lizard neoplasia case searches into Microsoft Excel (Microsoft 365 MSO (Version 2307)). Collected publication information included PubMed identification number/CAB ID code/Web of Science ID, title, authors, year of publication, and journal of publication. Collected animal information included the common and scientific name of the animal, age (in months), maturity (adult vs. juvenile (<3 months old)), sex (if known), body area/system of the primary mass, the histological diagnosis of the primary mass, primary mass malignancy, presence and location of any metastases, presence and identity of concurrent but non-metastatic masses, if treatment was pursued, treatment method pursued, case outcome, and survival from time of diagnosis (in months).

Diagnoses were grouped by primary histologic diagnosis, as reported by the primary investigator. The designation of benign versus malignant was primarily based on the reported diagnosis. Lymphoma and leukemia cases were categorically defined as malignant. In cases in which the reported diagnosis was ambiguous, neoplasms were classified as “malignant” in cases with observed distal (metastatic disease) or local invasion with tissue destruction [12]. If neither the diagnosis nor the description were sufficient, the neoplasm was classified as “undetermined.” Treatments were categorized as follows: surgery only (including complete and marginal excision), chemotherapy only, radiation therapy only (including external beam and strontium techniques), surgery and chemotherapy, surgery and radiation, unknown treatment, and no treatment provided. The diagnostic sampling of masses such as fine needle aspirates and incision biopsies was not categorized as a treatment.

#### 2.1.2. ESCRA Database Review

The ESCRA database was searched for submitted lizard neoplasia cases. Cases were confirmed to be neoplasia by board-certified veterinary pathologists working at either independent laboratories or university settings before entry per ESCRA submission guidelines.

Data were summarized from these lizard neoplasia case searches using Microsoft Excel (Microsoft 365 MSO (Version 2307)). The same animal data were collected from each submitted case as from the literature review. Diagnoses were grouped also by primary histologic diagnosis and behavior as reported by submitting clinicians and the benign/malignancy parameters applied to the literature review.

Treatments were categorized into groups per the ESCRA submission form as follows: surgery only (which included surgical excision, cryosurgery, or cryotherapy), chemotherapy only (which included electrochemotherapy, steroid therapies, and NSAIDs), radiation therapy only (including photoradiation and phototherapy), surgery and either radiation therapy or chemotherapy, surgery with radiation therapy and chemotherapy, supportive care (which included systemic or topical antibiotics and NSAIDs), unknown treatment, and no treatment provided. The diagnostic sampling of masses such as fine needle aspirates and incision biopsies were not categorized as treatments.

### 2.2. Data Analysis

Identified case information was analyzed using IBM Corp. IBM SPSS Statistics for Windows (released 2019; Version 26.0, Armonk, NY, USA) and R statistical computing software (version 4.3.2; R Core Team 2023. R: A language and environment for statistical computing. R foundation for Statistical Computing, Vienna, Austria, https://www.Rproject.org/, accessed on 7 December 2023) [13]. Boosting analysis was used to evaluate the data. For the statistical method of boosting, the mboost package (Model-based boosting, R package version 2.9-9, https://CRAN.R-project.org/package=mboost, accessed on 7 December 2023) was used. Neoplasms were grouped into more general classification categories by behavior (benign vs. malignant) as well as by tissue and tumor type for the mboost evaluation of the survival of animals affected by these types of neoplasms. For boosting analysis, predictor variables were modeled with the use of brandom as a base-learner. For each data set, a set of predictor variables was utilized. The literature data and ESCRA data sets were evaluated together and also separately using the eight variables of species, sex, life stage (juvenile or adult), primary neoplasia diagnosis, neoplasia malignancy, presence of metastasis, tumor location, and treatment type. Outcomes were assigned to lizards, with “1” designated for those that died or were euthanized due to the neoplasm and “0” designated for those that died due to another cause. Lizards without a known cause of death were included in the final analysis. The modeled effects of the predictor variables with outcomes for each animal evaluated were compared to a set of 2000 null model distributions of effects generated from modeling performed with permutated outcomes, with *p* < 0.05 being the threshold of significance regarding the tails of the null model distribution (two-sided hypothesis), similar in method to Mayr et al. [13]. Significance was calculated only for variables with two or more individuals in the represented population.

## 3. Results

### 3.1. Lizard Population

There were 274 individual lizards included in this study. There were 219 individual lizard cases identified in the literature review. The described search criteria applied to the ESCRA database yielded a total of 55 unique individual lizard cases. The individual cases represented 65 different species, with 59 species identified in the literature review, 20 species identified in the ESCRA database, and 14 species being common between the two.

#### 3.1.1. Literature

Green iguanas (*Iguana iguana*, *n* = 55) were the most represented species, followed by central bearded dragons (*Pogona vitticeps*, *n* = 32) and North African spiny-tailed lizard (*Varanus exanthematicus*, *n* = 10) (Table 1). Males (*n* = 40) were slightly more represented than females (*n* = 41), with the majority of the animals’ sex being unknown (*n* = 138). Life stage was generally unknown (*n* = 141), with adults (*n* = 52) being more represented than juveniles (*n* = 6). Individual age ranged from 5 months to 300 months (12 years) where age was available (*n* = 50), with a mean of 77.7 months (6.5 years) and a median of 60 months (5 years). Age information was not available for the remaining 169 individuals.

#### 3.1.2. ESCRA

Central bearded dragons (*Pogona vitticeps*, *n* = 25) were the most represented species, followed by Panther chameleons (*Furcifer pardalis*, *n* = 4) and Leopard geckos (*Eublepharis macularius*, *n* = 4) (Table 1). Males (*n* = 26) were more represented than females (*n* = 20), and a minority of individuals’ sex were unknown (*n* = 9). All individuals were either adults (*n* = 42) or their life stage was unknown (*n* = 13). Individual age ranged from 24 (2 years) to 173 months (14.4 years) for individuals where age was available (*n* = 34), with a median age of 72 months (6 years) and mean age of 81.5 months (6.8 years). Age information was not available for the remaining 21 individuals.

### 3.2. Neoplasia Information

There were 108 neoplasms included in this study. A total of 85 neoplasms were identified in the literature review sources, and 38 neoplasms were identified in the ESCRA database cases. Sixteen neoplasms were common between the two sources.

#### 3.2.1. Literature

Lymphoma was the most represented neoplasia (*n* = 18) followed by squamous cell carcinoma (*n* = 13), chromatophoroma (*n* = 10), and teratoma (*n* = 10) (Table 2). Malignant neoplasia (*n* = 133) was more represented than benign neoplasia (*n* = 58). Some cases (*n* = 28) did not have an identified location of neoplasia. The liver was the most commonly identified site (*n* = 15). An unknown presence of metastasis (*n* = 156) was more common than confirmed presence of metastasis (*n* = 13) or lack of metastasis (*n* = 50).

#### 3.2.2. ESCRA

Squamous cell carcinoma was the most represented neoplasia (*n* = 8) followed by spindle cell sarcoma (*n* = 4) (Table 3). Skin (*n* = 6) and skin/scales of the head region (*n* = 6) were the most commonly identified sites of neoplasia. Malignant neoplasia (*n* = 40) was more represented than benign neoplasia (*n* = 7), with many other cases not having identified malignancy information (*n* = 9). A lack of metastasis (*n* = 7) was slightly more common than the presence of metastasis (*n* = 5), with an unknown presence of metastasis being most common (*n* = 43).

### 3.3. Treatment Information

#### 3.3.1. Literature

Treatment was more commonly not pursued (*n* = 161) than pursued (*n* = 31), with surgery being the most common treatment performed (*n* = 25) followed by chemotherapy alone (*n* = 3), radiation therapy alone (*n* = 2), and surgery with chemotherapy (*n* = 1) (Table 4).

#### 3.3.2. ESCRA

The pursuit of treatment was either unknown or not attempted for the majority of cases (*n* = 36), compared to instances of treatment (*n* = 19). Surgery was the most common treatment performed (*n* = 17), followed by chemotherapy alone (*n* = 1) and surgery combined with radiation therapy (*n* = 1) (Table 5).

### 3.4. Case Outcome and Survival Times

#### 3.4.1. Literature

Case outcome information was available for 89 of the identified cases. Death or euthanasia due to neoplasia (*n* = 25) and the individual still being alive (*n* = 25) were the most commonly known outcomes. Death due to unknown causes (*n* = 11) and death not due to neoplasia (*n* = 1) were also identified. Case outcomes were unknown for the majority of cases (*n* = 130). Survival time information was available for 108 of the identified cases, ranging from 0 months to 56 months (4.6 years) after the time of diagnosis with neoplasia until death. Mean survival time was 20.0 months (1.7 years), with a median survival time of 0 months.

#### 3.4.2. ESCRA

Case outcome information was available for 41 of the identified cases. Death or euthanasia due to neoplasia was the most common outcome identified (*n* = 17), with death or euthanasia due to non-neoplastic causes (*n* = 14) and unknown status of survival (*n* = 14) as the next largest groups, followed by the individuals still being alive (*n* = 9). Only 1 individual was identified to have died from unknown causes. Survival time information was available for 13 of the identified cases, ranging from 0 months to 30 months (2.5 years) after the time of diagnosis with neoplasia until death. Mean survival time was 10.3 months, with a median survival time of 2 months.

### 3.5. Prognosis

The significance of predictor variables on individual outcomes was estimated using boosting and permutation as described above and applied to each of the literature and ESCRA sample populations. This provided estimations for the impact of a given variable on whether an individual case would result in death from diagnosed neoplasia or survival or death from other causes (referred to as “non-neoplastic outcomes” in this paper).

#### 3.5.1. Combined Data from Literature and ESCRA

Komodo dragons and savannah monitors were the species identified as being significantly more likely to die due to neoplasia (*p* < 0.05). Panther chameleons and round island skinks were significantly associated with non-neoplastic outcomes of death (*p* < 0.05). Leukemia and islet cell carcinoma were significantly associated with death due to neoplasia (*p* < 0.05). Chondrosarcoma, myxosarcoma, osteosarcoma, and squamous cell carcinoma were significantly associated with non-neoplastic outcomes (*p* < 0.05). Surgery alone and radiation therapy alone each were significantly associated with non-neoplastic outcomes, while lizards not receiving treatment were significantly associated with death due to neoplasia (*p* < 0.05). Unknown sex was significantly associated with non-neoplastic outcomes of death (*p* < 0.05), whereas unknown life stage was significantly associated with death due to neoplasia (*p* < 0.05). Benign neoplasia was significantly associated with non-neoplastic outcomes of death (*p* < 0.05), and an unknown status of being benign or malignant was associated with death due to neoplasia (*p* < 0.05). The tumor location of the liver was significantly associated with death due to neoplasia, while some of the other tumor locations were significantly associated with non-neoplastic outcomes of death (*p* < 0.05) (Table 6). A contrast in survival times relating to death being due to neoplasia (with or without treatment) compared to non-neoplastic outcomes is shown in Figure 1. For both benign and malignant neoplasia, neoplasia without treatment exhibits a faster rate of mortality in comparison to neoplasia with treatment. Malignant neoplasia also exhibits a faster rate of mortality in comparison to benign neoplasia. For malignant neoplasia, non-neoplastic outcomes of death were intermediate in the rate of mortality compared to neoplasia without treatment and neoplasia with treatment.

#### 3.5.2. Literature

Malignant neoplasia was significantly associated with death due to neoplasia (*p* < 0.05). Radiation alone and surgery alone were significantly associated (*p* < 0.05) with non-neoplastic outcomes of death, while no treatment was significantly associated with death due to neoplasia (*p* < 0.05). Chondrosarcoma, osteosarcoma, and squamous cell carcinoma were significantly associated with non-neoplastic outcomes (*p* < 0.05), while papilloma was significantly associated with death due to neoplasia (*p* < 0.05). Tumor location in the hematopoietic/lymphatic system was significantly associated with death (*p* < 0.05). No metastasis was significantly associated with death due to neoplasia (*p* < 0.05), while unknown sex was significantly associated with non-neoplastic outcomes of death (Table 7). Round island skinks were significantly associated with non-neoplastic outcomes of death (*p* < 0.05), while savannah monitors were significantly associated with death due to neoplasia (*p* < 0.05). Jaw and skin tumor locations were significantly associated with non-neoplastic outcomes of death (*p* < 0.05), while bone marrow and liver tumor locations were significantly associated with death due to neoplasia (*p* < 0.05).

#### 3.5.3. ESCRA

Cases where the status of metastasis was unknown or the life stage (juvenile or adult) was unknown were significantly more likely to die due to non-neoplastic causes of death (*p* < 0.05). Benign neoplasia and the adult life stage were significantly associated (*p* < 0.05) with death due to non-neoplastic causes of death. Papilloma was unexpectedly found to be significantly associated with death due to neoplasia (*p* < 0.05). Surgery alone was significantly associated (*p* < 0.05) with death due to non-neoplastic causes, while no treatment was significantly associated with death due to neoplasia (*p* < 0.05). Tumor location in the gastrointestinal/hepatic system was significantly associated with death due to neoplasia, while tumor location in the gastrointestinal and alimentary parts of the abdominal organ were significantly associated with death due to neoplasia (*p* < 0.05) (Table 8).

## 4. Discussion

Exotic animal oncology is still a developing field of study within veterinary medicine with large gaps in the understanding of disease progression, prognosis, and treatment [113]. This lack of understanding of exotic animal oncology is partly due to a lack of research focused on characterizing disease in these animals. This issue results in a large amount of extrapolation from the understanding of similar neoplastic diseases in cat, dog, and human models. The purpose of this study was to increase the knowledge of neoplasia in lizards specifically and to evaluate the effects of neoplasia on individual animals by exploring the relationships between different variables and patient outcomes. This information could be used to aid in the management of similar clinical cases by providing insight into factors that may positively or negatively affect a patient’s ultimate outcome.

Through this study, we sought to evaluate neoplasia in lizards by investigating the literature as well as a database mainly containing unpublished data on tumor cases. The combination of these two data sources produced a broader range of predictions than either data source analyzed alone. Our approach in using both ESCRA and published datasets for the analyses therefore provided a more informative evaluation of the clinical treatment of neoplasia in lizards.

Consensus between predictions stemming from each separate data source were limited and were specific to negative prognoses for cases having no treatment attempted and positive prognoses for cases involving having surgery only. ESCRA uses a standardized approach to collecting relevant clinical information. Unfortunately, a similar guideline for published data in the literature does not exist and would have improved comparisons. The literature was also substantially affected by the publication bias of unique cases, first-reported treatments, or first-reported deaths. Another issue impacting consensus may relate to how it was determined if a patient died due to underlying neoplasia or another cause, which is reliant on the interpretation of the treating veterinarian, pathologist, or researcher.

It does appear that the measurement of survival times is useful in interpreting the association of treatment with neoplasia prognoses. Further investigation would ideally collect information on tumor response (size over time) and other time series data, which would lead to better information regarding clinical intervention strategies for neoplasia in lizards.

While evaluating the species represented in this study, two species (Komodo dragons and savannah monitors) were identified as being statistically more likely to die due to neoplasia than non-neoplastic outcomes. Conversely, panther chameleons and round island skinks were identified as being statistically more likely to die due to non-neoplastic outcomes than neoplasia. These species were not among the most represented species within the literature data, with representation being at *n* = 5 or less, so additional prevalence or incidence investigation is needed to determine if each finding is a true species difference, influenced by limited sample size, or a publication bias.

Of the neoplasms identified in this study, islet cell carcinoma and leukemia cases within the combined data of the literature and ESCRA data sources were associated with death due to neoplasia. Papilloma cases were also associated with death due to neoplasia, but only within the literature data source. Chondrosarcoma, myxosarcoma, osteosarcoma, and squamous cell carcinoma cases from the combined data of the ESCRA and literature data sources were found to be significantly associated with non-neoplastic outcomes of death. The findings concerning papillomas were unexpected due to the typical behavior of the neoplasms. Possible explanations for these findings may be advanced life stage, concurrent systemic disease, owner decisions to pursue euthanasia, excessive size of the neoplasm that inhibited movement, eating or normal life actions, or other complications that lead to death due to other reasons. Additional investigation should be undertaken to confirm these findings, as a better understanding of the general progression and severity of a disease and the effectiveness of treatment is of great clinical importance. Compared to this, squamous cell carcinoma cases were found to have the greatest association with non-neoplastic outcomes. This neoplasia, while typically significantly locally aggressive, can often be managed with adjunctive therapies that allow an animal to survive long enough to die from non-neoplastic causes [114] and may represent a true association.

Of the combined set of 274 cases, metastasis was reported for only 6.6% of these cases. The presence of metastasis was unknown in the majority of cases evaluated (73%). The reason for having a majority of cases where metastasis data is not reported is most likely due to a lack of thorough post-mortem testing on all tissues in these retrospective cases to confirm or deny the presence of metastasis. The encouragement of full necropsies of representative tissues will improve the future evaluation of lizard neoplasia cases to assess metastasis. Benign neoplasia was expectedly associated significantly with outcomes of survival or death due to non-neoplastic causes in the ESCRA data set. Malignant neoplasia was significantly associated with death due to neoplasia in the literature data set. Generally, treatment was not sought in the majority of all cases (81%). A lack of treatment was found to significantly negatively impact patient outcomes. Multiple factors can play into the decision to seek treatment or not, including the type of neoplasm present, the current condition of the patient, the availability of different treatments, and personal factors for the owner. It is not unexpected that the lack of treatment would lead to euthanasia or the death of the animal due to the underlying neoplasia. Without addressing the underlying disease process, most neoplasms will negatively affect the quality of life and lifespan of an individual over time.

Sex as a variable was not found to have a significant association with case outcome, but cases where sex was not reported were found to be significantly associated with death due to non-neoplastic causes. In reviewing the cases of individuals with unknown sex included in this study, it was observed that the majority of cases also experienced a lack of treatment by the owners, which, as discussed above, was shown to be significantly associated with neoplasia-related death. This may reflect a lack of investment in the animals by the owners or could be a reflection of the significant effect of not seeking treatment for these conditions. Regardless, these animals appeared to survive or to die of non-neoplastic causes, and ongoing investigation and statistical analysis is indicated to confirm this association. The surgical removal of neoplasia had a significant association with averting death due to neoplasia and had a greater effect in this regard than treatment with radiation therapy alone. This is consistent with previous publications investigating neoplasia in lizards [1,113]. Surgical excision is the basis of most oncological treatment procedures, and even if no further treatment is performed, it is known to prolong the duration and quality of life in veterinary patients. Surgical excision gives a chance for the complete removal of the neoplasia, through which the best prognosis can be achieved [115]. Treatment protocols remain a developing aspect of exotic animal oncology, and precise dosing and choice of medications are still being established [14,113]. An even greater range of positive outcomes in patients receiving treatment of different modalities may result once more refined protocols are created and used with greater frequency.

## 5. Conclusions

Multiple predictor variables were significantly associated with death or euthanasia due to neoplasia. Some of the greatest effects in this association were for lizards not receiving treatment, tumors of the liver, leukemia, and tumors for which the statuses of life stage and of being benign versus malignant were unknown. Conversely, the greatest effects of predictor variable values for lizards that did not die due to neoplasia were due to those receiving radiation alone, those with benign tumors, squamous cell carcinoma tumor types, and tumors of the skin/scales head region. In general, treatment for neoplasia contributed to longer survival times in contrast to the rapid rate of death associated with animals dying of neoplasia that did not receive treatment.

## Figures and Tables

**Figure 1 animals-14-01395-f001:**
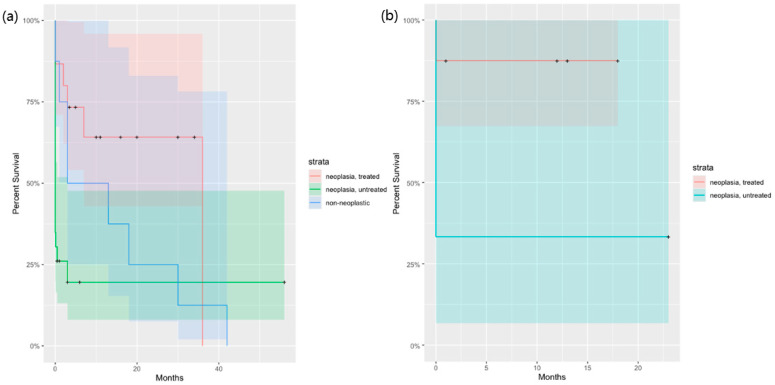
Kaplan–Meier survival plot of survival percentages according to status of neoplasia and treatment. + symbols are for right-censored data (i.e., still alive). Shaded areas show 95% confidence intervals. (**a**) Cases of malignant neoplasia in lizards, including those dying from non-neoplastic cause. (**b**) Cases of benign neoplasia in lizards.

**Table 1 animals-14-01395-t001:** Frequency of the most prevalent lizard species with neoplasia from the literature and ESCRA sources.

Literature Species (Scientific Name)	Frequency	ESCRA Species (Scientific Name)	Frequency
Beaded lizard (*Heloderma exasperatum/horridum*)	2/219 (0.91%)	Central bearded dragon (*Pogona vitticeps*)	21/55 (45.5%)
Broad headed skink *(Eumeces laticeps*)	2/219 (0.91%)	Blotched blue-tongued lizard (*Tiliqua nigrolutea*)	2/55 (3.6%)
Central bearded dragon (*Pogona vitticeps*)	32/219 (14.6%)	Green iguana (*Iguana iguana*)	3/55 (5.5%)
Crocodile lizard (*Shinisaurus crocodilurus)*	2/219 (0.91%)	Leopard gecko (*Eublepharis macularius*)	4/55 (7.3%)
Desert grassland whiptail lizard (*Cnemidophorus uniparens*)	2/219 (0.91%)	Panther chameleon (*Furcifer pardalis*)	4/55 (7.3%)
East Indian water lizard (*Hydrosaurus amboinensis*)	2/219 (0.91%)	Veiled chameleon (*Chamaeleo calyptratus*)	3/55 (5.5%)
European green lizard (*Lacerta viridis*)	4/219 (1.8%)		
Gila monster (*Heloderma suspectum*)	5/219 (2.3%)		
Green anole (*Anolis carolinensis*)	2/219 (0.91%)		
Green iguana (*Iguana iguana*)	55/219 (28%)		
Komodo dragon (*Varanus komodoensis*)	3/219 (1.4%)		
Leopard gecko (*Eublepharis macularius*)	5/219 (2.3%)		
Mexican beaded lizard (*Heloderma horridum*)	2/219 (0.91%)		
Panther chameleon (*Furcifer pardalis*)	2/219 (0.91%)		
Round island skink (*Leiolopisma telfairii*)	2/219 (0.91%)		
Savannah monitor (*Varanus exanthematicus*)	10/219 (4.6%)		
Spiny-tailed lizard (*Uromastyx acanthinura*)	10/219 (4.6%)		
Spiny-tailed monitor (*Varanus acanthurus*)	3/219 (1.4%)		
Veiled chameleon (*Chamaeleo calyptratus*)	8/219 (3.7%)		

**Table 2 animals-14-01395-t002:** Frequency of the most prevalent neoplasias from reviewed literature sourced with identified affected lizard species, malignancy behavior, and represented individuals reported.

Neoplasia (Malignancy Behavior)	Frequency	Affected Species (Scientific Name)	Represented Individuals
Biliary adenocarcinoma (malignant)	6/219 (2.7%)	Broad headed skink (*Eumeces laticeps*)	1
		Green iguana (*Iguana iguana*)	3
		Plumed basilisk (*Basiliscus plumifrons*)	1
		Spiny tailed iguana (*Ctenosaura pectinate*)	1
Carcinoma (malignant)	3/219 (1.4%)	Central bearded dragon (*Pogona vitticeps*)	2
		Warren’s girdled lizard (*Cordylus warren*)	1
Cholangiocarcinoma (malignant)	4/219 (1.8%)	Central bearded dragon (*Pogona vitticeps*)	1
		Green iguana (*Iguana iguana*)	2
		Texas horned lizard (*Phrynosoma cornutum*)	1
Chondrosarcoma (malignant)	2/219 (0.91%)	Spiny-tailed monitor (*Varanus acanthurus*)	1
		Storr’s monitor *(Varanus storri*)	1
Chromatophoroma (benign)	3/219 (1.6%)	Green iguana (*Iguana iguana*)	1
		Savannah monitor (*Varanus exanthematicus*)	1
		Veiled chameleon (*Chamaeleo calyptratus*)	1
Chromatophoroma (malignant)	6/219 (2.7%)	Central bearded dragon (*Pogona vitticeps*)	1
		Day Gecko (*Gekko phelsuma*)	1
		Dwarf bearded dragon (*Pogona henrylawsoni*)	1
		Leopard gecko (*Eublepharis macularius*)	1
		Veiled chameleon (*Chamaeleo calyptratus*)	2
Colon adenocarcinoma (malignant)	2/219 (0.91%)	Leopard gecko (*Eublepharis macularius*)	1
		Mexican beaded lizard (*Heloderma horridum*)	1
Fibroma (benign)	6/219 (2.7%)	Green iguana (*Iguana iguana*)	4
		Veiled chameleon (*Chamaeleo calyptratus*)	2
Fibropapilloma (benign)	2/219 (0.91%)	Green iguana (*Iguana iguana*)	1
		Sailfin lizard (*Hydrosaurus postulatus*)	1
Fibrosarcoma (malignant)	3/219 (1.4%)	Central bearded dragon (*Pogona vitticeps*)	1
		Spiny-tailed lizard (*Ctenosaura pectinata*)	1
		Unknown water dragon (*Physignathus* sp.)	1
Gastric carcinoid (malignant)	3/219 (1.4%)	Central bearded dragon (*Pogona vitticeps*)	3
Granulosa cell tumor (malignant)	3/219 (1.4%)	Green iguana (*Iguana iguana*)	2
		Savannah monitor (*Varanus exanthematicus*)	1
Hemangiosarcoma (malignant)	2/219 (0.91%)	Green iguana (*Iguana iguana*)	1
		Hispaniolan curly-tailed lizard (*Leiocephalus schreibersii*)	1
Hepatocellular carcinoma (malignant)	3/219 (1.4%)	Green iguana (*Iguana iguana*)	4
Hepatoma (benign)	2/219 (0.91%)	Broad headed lizard (*Eumeces laticeps*)	1
		Gila monster (*Heloderma suspectum*)	1
Interstitial cell adenoma (benign)	2/219 (0.91%)	Cuban iguana (*Cyclura nubila*)	1
		Green iguana (*Iguana iguana*)	1
Islet cell carcinoma (malignant)	3/219 (1.4%)	Komodo dragon (*Varanus komodoensis*)	3
Leiomyosarcoma (malignant)	5/219 (1.4%)	Central bearded dragon (*Pogona vitticeps*)	3
Leukemia (malignant)	5/219 (2.3%)	Central bearded dragon (*Pogona vitticeps*)	2
		Green iguana (*Iguana iguana*)	2
		Green tree monitor (*Varanus prasinus*)	1
Lymphoma (malignant)	18/219 (8.2%)	East Indian water lizard (*Hydrosaurus amboinensis*)	1
		Green iguana (*Iguana iguana*)	5
		Italian wall lizard (*Lacerta sicula*)	1
		Savannah monitor (*Varanus exanthematicus*)	3
		Spiny-tailed lizard (*Uromastyx acanthinura*)	8
Melanophoroma (benign)	2/219 (0.91%)	Green iguana (*Iguana iguana*)	2
Melanophoroma (malignant)	3/219 (1.4%)	Beaded lizard	1
		Leopard gecko (*Eublepharis macularius*)	1
		Veiled chameleon (*Chamaeleo calyptratus*)	1
Myxoma (benign)	3/219 (1.4%)	Central bearded dragon (*Pogona vitticeps*)	1
		Green iguana (*Iguana iguana*)	2
Osteosarcoma (malignant)	5/219 (2.3%)	Central bearded dragon (*Pogona vitticeps*)	1
		Green iguana (*Iguana iguana*)	2
		Spiny-tailed monitor (*Varanus acanthurus*)	2
Pancreatic adenocarcinoma (malignant)	2/219 (0.91%)	Solomon island skink (*Corucia zebrata*)	1
		Veiled chameleon (*Chamaeleo calyptratus*)	1
Papilloma (benign)	3/219 (1.4%)	Ocellated lizard (*Lacerta lepida*)	1
		European green lizard (*Lacerta viridis*)	2
Parathyroid adenoma (benign)	4/219 (1.8%)	Green iguana (*Iguana iguana*)	5
Renal adenoma (benign)	2/219 (0.91%)	Green iguana (*Iguana iguana*)	2
Squamous cell carcinoma (malignant)	13/219 (5.9%)	Blue-tongued skink (*Tiliqua scincoides*)	1
		Central bearded dragon (*Pogona vitticeps*)	4
		European green lizard (*Lacerta viridis*)	1
		Leopard gecko (*Eublepharis macularius*)	2
		Panther chameleon (*Furcifer pardalis*)	2
		Round island skink (*Leiolopisma telfairii*)	2
		Spiny-tailed monitor (*Varanus acanthurus*)	1
Teratoma (benign)	7/219 (3.2%)	Central bearded dragon (*Pogona vitticeps*)	1
		Desert grassland whiptail lizard (*Cnemidophorus uniparens*)	2
		Green iguana (*Iguana iguana*)	3
		Lau banded iguana (*Brachylophus fasciatus*)	1
Teratoma (malignant)	3/219 (1.4%)	Central bearded dragon (*Pogona vitticeps*)	1
		Green iguana (*Iguana iguana*)	2
Thyroid carcinoma (malignant)	3/219 (1.4%)	Marbled velvet gecko (*Oedura marmorata*)	1
		Centralian rough-knob tail gecko (*Nephrusus amyae*)	1
		Smooth knob-tail gecko (*Nephrusus levis*)	1
Undifferentiated sarcoma (malignant)	2/219 (0.091%)	Chuckwalla (*Sauromalus obesus*)	1
		Five-lined skink (*Eumeces fasciatus*)	1

[1,2,3,4,5,6,7,8,9,10,11,14,15,16,17,18,19,20,21,22,23,24,25,26,27,28,29,30,31,32,33,34,35,36,37,38,39,40,41,42,43,44,45,46,47,48,49,50,51,52,53,54,55,56,57,58,59,60,61,62,63,64,65,66,67,68,69,70,71,72,73,74,75,76,77,78,79,80,81,82,83,84,85,86,87,88,89,90,91,92,93,94,95,96,97,98,99,100,101,102,103,104,105,106,107,108,109,110,111,112].

**Table 3 animals-14-01395-t003:** Frequency of the most prevalent neoplasms from the ESCRA database with identified affected lizard species, malignancy behavior, and represented individuals reported.

Neoplasia (Malignancy Behavior)	Frequency	Affected Species (Scientific Name)	Represented Individuals
Chromatophoroma (benign)	1/55 (1.8%)	Veiled chameleon (*Chamaeleo calyptratus*)	1
Chromatophoroma (malignant)	1/55 (1.8%)	Central bearded dragon (*Pogona vitticeps*)	1
Leukemia (malignant)	2/55 (3.6%)	Central bearded dragon (*Pogona vitticeps*)	2
Lymphosarcoma (malignant)	2/55 (3.6%)	Central bearded dragon (*Pogona vitticeps*)	1
		Western banded gecko (*Coleonyx variegatus*)	1
Myxosarcoma (malignant)	2/55 (3.6%)	Central bearded dragon (*Pogona vitticeps*)	2
Papilloma (benign)	1/55 (1.8%)	Green iguana (*Iguana iguana*)	1
Papilloma (unknown)	1/55 (1.8%)	Veiled chameleon (*Chamaeleo calyptratus*)	1
Sarcoma (malignant)	1/55 (1.8%)	Central bearded dragon (*Pogona vitticeps*)	1
Sarcoma (unknown)	1/55 (1.8%)	Central bearded dragon (*Pogona vitticeps*)	1
Spindle cell sarcoma (malignant)	4/55 (7.3%)	Blotched blue-tongued lizard (*Tiliqua nigrolutea*)	1
		Central bearded dragon (*Pogona vitticeps*)	2
		Leopard gecko (*Eublepharis macularius*)	1
Squamous cell carcinoma (malignant)	7/55 (12.7%)	Central bearded dragon (*Pogona vitticeps*)	4
		Leopard gecko (*Eublepharis macularius*)	1
		Panther chameleon (*Furcifer pardalis*)	2

**Table 4 animals-14-01395-t004:** Frequency of the most prevalent treatments sought for lizards with neoplasia from literature sources with average survival time, species, and identified neoplasia reported.

Treatment	Frequency	Average Survival Time (Months)	Species (Scientific Name)	Neoplasia
Chemotherapy only	3/219 (1.4%)	3.5	Central bearded dragon (*Pogona vitticeps*)	Squamous cell carcinoma
			Emerald monitor (*Varanus prasinus*)	Leukemia
Radiation only	2/219 (0.91%)	19.5	Central bearded dragon (*Pogona vitticeps*)	Myxosarcoma
			Green iguana (*Iguana iguana*)	Lymphoma
Surgery only	25/219 (11.4%)	10.4	Beaded lizard (*Heloderma horridum*)	Adenocarcinoma
			Blue-tongued skink (*Tiliqua scincoides*)	Squamous cell carcinoma
			Central bearded dragon (*Pogona vitticeps*)	Carcinoma, anaplastic sarcoma, myxoma, peripheral nerve sheath tumor, teratoma, squamous cell carcinoma
			Day Gecko (*Gekko phelsuma*)	Chromatophoroma
			European Green lizard (*Lacerta viridis*)	Papilloma
			Green iguana (*Iguana iguana*)	Adenoma, benign melanophoroma, chromatophoroma, granulosa cell tumor, adrenal cortical adenocarcinoma, adrenal cortical adenocarcinomas, multifocalc cholangiocarcinoma, teratoma
			Lau banded iguana (*Brachylophus fasciatus*)	Teratoma
			Leopard gecko (*Eublepharis macularius*)	Chromatophoroma
			Marbled velvet gecko (*Oedura marmorata*)	Thyroid carcinoma
			Panther chameleon (*Furcifer pardalis*)	Squamous cell carcinoma
			Savannah monitor (*Varanus exanthematicus*)	Mesothelioma
			Veiled chameleon (*Chamaeleo calyptratus*)	Chromatophoroma

**Table 5 animals-14-01395-t005:** Frequency of the most prevalent treatments sought from the ESCRA database of lizards with neoplasia with average survival time, species, and identified neoplasia reported.

Treatment	Frequency	Average Survival Time (Months)	Species (Scientific Name)	Neoplasia
Chemotherapy only	1/55 (1.8%)	Not reported	Central bearded dragon (*Pogona vitticeps*)	Leukemia
Surgery and radiation	1/55 (1.8%)	Not reported	Central bearded dragon (*Pogona vitticeps*)	Anaplastic sarcoma
Surgery only	17/55 (30.9%)	9.75	Central bearded dragon (*Pogona vitticeps*)	Fibrosarcoma, myxosarcoma, sarcoma, spindle cell sarcoma, squamous cell carcinoma, chondrosarcoma
			Fat-tailed gecko (*Hemitheconyx caudicinctus*)	Neurofibrosarcoma
			Green anole (*Anolis carolinensis*)	Carcinoma
			Green iguana (*Iguana iguana*)	Sertoli cell tumor
			Leopard gecko (*Eublepharis macularius*)	Squamous cell carcinoma
			Panther chameleon (*Furcifer pardalis*)	Squamous cell carcinoma, papilloma
			Veiled chameleon (*Chamaeleo calyptratus*)	Chromatophoroma

**Table 6 animals-14-01395-t006:** Predictor variable values of lizards with neoplasia that had significant association with treatment outcomes based on boosted modeling from the combined data of reviewed literature sources and the ESCRA database.

Predictor Variable	Variable Value	Sample Size	Outcome ^a^
Benign vs. malignant	Benign	20	0.10 ^b^
	Unknown	6	−0.073 ^b^
Histological diagnosis	Chondrosarcoma	2	0.028 ^b^
	Islet cell carcinoma	3	−0.038 ^b^
	Leukemia	4	−0.060 ^b^
	Myxosarcoma	2	0.027 ^b^
	Osteosarcoma	3	0.040 ^b^
	Squamous cell carcinoma	3	0.12 ^b^
Life stage	Unknown	7	−0.077 ^b^
Sex	Unknown	17	0.074 ^b^
Species	Komodo dragon *(Varanus komodoensis)*	3	−0.038 ^b^
	Panther chameleon *(Furcifer pardalis)*	4	0.046 ^b^
	Round island skink *(Leiolopisma telfairii)*	2	0.025 ^b^
	Savannah monitor (*Varanus exanthematicus*)	5	−0.070 ^b^
	Spiny-tailed monitor *(Varanus acanthurus)*	2	0.027 ^b^
Treatment	Surgery alone	35	0.12 ^b^
	Radiation alone	2	0.031 ^b^
	No treatment	52	−0.13 ^b^
Tumor location	Abdominal organ (gastrointestinal and alimentary)	3	0.046
	Arm/leg (joint/bone/cartilage)	2	0.028 ^b^
	Jaw	2	0.027 ^b^
	Kidney	2	0.031 ^b^
	Liver	4	−0.053 ^b^
	Pancreas	3	0.042 ^b^
	Skin/scales (head region)	4	0.049 ^b^

^a^ Outcome effect is relative to positive or negative outcomes with each clinical report scored as +1 and 0, respectively. Negative outcome is death due to neoplasia, and positive outcome is non-neoplastic (survival or death to non-neoplastic cause). ^b^ Significance of outcome effect based on *p* < 0.05.

**Table 7 animals-14-01395-t007:** Predictor variable values of lizards with neoplasia that had significant association with treatment outcomes based on boosted modeling from the reviewed literature sources.

Predictor Variable	Variable Value	Sample Size	Outcome ^a^
Benign vs. malignant	Malignant	41	−0.076 ^b^
Histological diagnosis	Chondrosarcoma	2	0.046 ^b^
	Osteosarcoma	3	0.062 ^b^
	Papilloma	2	−0.068 ^b^
	Squamous cell carcinoma	7	0.14 ^b^
Metastasis	No	39	−0.077 ^b^
Sex	Unknown	16	0.095 ^b^
Species	Round island skink (*Leiolopisma telfairii)*	2	0.039 ^b^
	Savannah monitor (*Varanus exanthematicus*)	4	−0.083 ^b^
Treatment	Radiation only	2	0.054 ^b^
	Surgery only	20	0.095 ^b^
	No treatment	29	−0.14 ^b^
Tumor location	Bone marrow	3	−0.069 ^b^
	Jaw	2	0.042 ^b^
	Liver	4	−0.079 ^b^
	Skin	5	0.093 ^b^

^a^ Outcome effect is relative to positive or negative outcomes with each clinical report scored as +1 and 0, respectively. Negative outcome is death due to neoplasia, and positive outcome is non-neoplastic (survival or death from non-neoplastic cause). ^b^ Significance of outcome effect based on *p* < 0.05.

**Table 8 animals-14-01395-t008:** Predictor variable values of lizards with neoplasia that had significant association with treatment outcomes based on boosted modeling from the ESCRA database.

Predictor Variable	Variable Value	Sample Size	Outcome ^a^
Benign vs. malignant	Benign	6	0.13
	Unknown	6	−0.13
Life stage	Adult	34	0.085 ^b^
	Unknown	6	−0.11 ^b^
Treatment	Surgery only	15	0.18 ^b^
	No treatment	23	−0.13 ^b^
Tumor location	Abdominal organ (gastrointestinal/alimentary)	5	−0.094 ^b^

^a^ Outcome effect is relative to positive or negative outcomes with each clinical report scored as +1 or 0, respectively. Negative outcome is death due to neoplasia, and positive outcome is non-neoplastic (survival or death due to non-neoplastic cause). ^b^ Significance of outcome effect based on *p* < 0.05.

## Data Availability

Deidentified data are available through an approved research request to the Exotic Species Cancer Research Alliance (https://escra.cvm.ncsu.edu/, accessed on 4 May 2024).

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
