# Peer review of "Evaluation of Neoplasia, Treatments, and Survival in Lizard Species"

_animals, 2024, doi:10.3390/ani14101395_

Round 1

Reviewer 1 Report

Comments and Suggestions for Authors

The manuscript "EVALUATION OF NEOPLASIA, TREATMENTS AND SURVIVAL IN LIZARD SPECIES" reviews neoplasia in lizards using the current literature and the Exotic Species Cancer Research Alliance (ESCRA) database. It aims to provide updated prognostic information. The sample size is small, but I understand the limitations of an undertaking like this.

I have edited the manuscript for grammar and clarity, which is included in the attached Word document. Some significant points that should be addressed before publication:

1. The paper reads like a statistics project rather than a tool for clinicians trying to understand neoplasia in reptiles. Statements like "was significantly associated with non-neoplasia-related outcomes of either survival or death" are really difficult to understand and apply. I strongly recommend reframing the results reported in these statements in a way that is more accessible.

2. Some of your results and conclusions directly oppose what is currently understood about reptile neoplasia. For example, you state that squamous cell carcinoma had a positive prognosis, and malignant neoplasia was not significantly associated with death due to neoplasia. I think it would be good to be more explicit that these findings need further investigation and may be a result of the reports included in this study.

Comments on the Quality of English Language

I have made suggestions on grammar and readability in the attached Word document.

Author Response

Open Review

Quality of English Language

( ) I am not qualified to assess the quality of English in this paper
( ) English very difficult to understand/incomprehensible
( ) Extensive editing of English language required
( ) Moderate editing of English language required
(x) Minor editing of English language required
( ) English language fine. No issues detected

Yes

Can be improved

Must be improved

Not applicable

Does the introduction provide sufficient background and include all relevant references?

(x)

( )

( )

( )

Are all the cited references relevant to the research?

(x)

( )

( )

( )

Is the research design appropriate?

(x)

( )

( )

( )

Are the methods adequately described?

(x)

( )

( )

( )

Are the results clearly presented?

( )

( )

(x)

( )

Are the conclusions supported by the results?

( )

( )

(x)

( )

Comments and Suggestions for Authors

The manuscript "EVALUATION OF NEOPLASIA, TREATMENTS AND SURVIVAL IN LIZARD SPECIES" reviews neoplasia in lizards using the current literature and the Exotic Species Cancer Research Alliance (ESCRA) database. It aims to provide updated prognostic information. The sample size is small, but I understand the limitations of an undertaking like this.

I have edited the manuscript for grammar and clarity, which is included in the attached Word document. Some significant points that should be addressed before publication:

  • Recommended grammar adjustments were incorporated into the document per the provided Word document.
  1. The paper reads like a statistics project rather than a tool for clinicians trying to understand neoplasia in reptiles. Statements like "was significantly associated with non-neoplasia-related outcomes of either survival or death" are really difficult to understand and apply. I strongly recommend reframing the results reported in these statements in a way that is more accessible.
  • The phrase "non-neoplasia-related outcomes of either survival or death" was changed to “non-neoplastic outcomes” with definition in the abstract and results sections explaining it to mean either survival of the animal or death due to other non-neoplastic causes.
  1. Some of your results and conclusions directly oppose what is currently understood about reptile neoplasia. For example, you state that squamous cell carcinoma had a positive prognosis, and malignant neoplasia was not significantly associated with death due to neoplasia. I think it would be good to be more explicit that these findings need further investigation and may be a result of the reports included in this study.
  • Thank you for this comment. In regards to squamous cell carcinoma, this results does generally line up with the clinical outcome of the neoplasm where while it is locally invasive, patients that are afflicted with it typically die or are euthanized of other causes rather than the neoplasia directly.
  • Malignant may be due to sample size and the unique aspects of the provided cases (ones that lived)

Reviewer 2 Report

Comments and Suggestions for Authors

This manuscript describes and attempts to compile a large amount of data from a diverse group of species and diagnoses. Given this diversity along with limited availability of treatment and survival data, loss to follow up, and lack of necropsies (as is typical in a large literature-based review), it is difficult to pull it all together and reach clinically significant conclusions. This report does include some relevant data, but also fairly frequently reports non-relevant data that is deemed to be statistically significant but lacks clinical significance (e.g. lack of data correlated with survival). Perhaps further clinically relevant outcomes could be reached if the data was further breakdown into diagnoses (at least benign versus malignant) or by species (or even by infraorder to increase numbers and attempt to find trends) would make it more relevant, especially for treatment and survival times.  

The following phrase ‘non- neoplasia-related outcomes of either survival or death’ is used frequently within the text and is a confusing way to convey the information. It could be re-phrased to more clearly state what is trying to be conveyed.  

While the search parameters for the literature review component were well described, this manuscript would be improved by including more recently published papers. A quick PubMed search for ‘lizard neoplasia’ reveals multiple case reports and case series of neoplasia in lizards that have been published since August 2021. It also brought up other case reports and a few case series that were not listed in the references that should be included in this report. Below are some examples, but this is an incomplete list.

Examples of pre-August 2021 papers, not cited:

·         Williams MJ, Wong HE, Priestnall SL, Szladovits B, Stapleton N, Hedley J. Anaplastic Sarcoma and Sertoli Cell Tumor in a Central Bearded Dragon (Pogona vitticeps). J Herpetol Med Surg. 2020 Jun 11;30(2):68-73. doi: 10.5818/18-04-154.1. PMID: 33633500; PMCID: PMC7116819.

·         Magnotti JM, Garner MM, Stahl SJ, Corbin EM, LaDouceur EEB. RETROSPECTIVE REVIEW OF HISTOLOGIC FINDINGS IN CAPTIVE GILA MONSTERS (HELODERMA SUSPECTUM) AND BEADED LIZARDS (HELODERMA HORRIDUM). J Zoo Wildl Med. 2021 Apr;52(1):166-175. doi: 10.1638/2020-0058. PMID: 33827173.

·         Schilliger L, Paillusseau C, Gandar F, De Fornel P. IRIDIUM 192 (192-Ir) HIGH DOSE RATE BRACHYTHERAPY IN A CENTRAL BEARDED DRAGON (POGONA VITTICEPS) WITH ROSTRAL SQUAMOUS CELL CARCINOMA. J Zoo Wildl Med. 2020 Mar 17;51(1):241-244. doi: 10.1638/2019-0077. PMID: 32212571.

·         Abou-Madi N, Kern TJ. Squamous cell carcinoma associated with a periorbital mass in a veiled chameleon (Chamaeleo calyptratus). Vet Ophthalmol. 2002 Sep;5(3):217-20. doi: 10.1046/j.1463-5224.2002.00244.x. PMID: 12236875.

·         Goe AM, Heard DJ, Abbott JR, de Mello Souza CH, Taylor KR, Sthay JN, Wellehan JF. Surgical management of an odontogenic tumor in a banded Gila monster (Heloderma suspectum cinctum) with a novel herpesvirus. Vet Q. 2016 Jun;36(2):109-14. doi: 10.1080/01652176.2016.1153169. Epub 2016 Mar 4. PMID: 26863112.

·         Mendyk RW, Newton AL, Baumer M. A retrospective study of mortality in varanid lizards (Reptilia: Squamata: Varanidae) at the Bronx Zoo: implications for husbandry and reproductive management in zoos. Zoo Biol. 2013 Mar;32(2):152-62. doi: 10.1002/zoo.21043. Epub 2012 Sep 19. PMID: 22997089.

Examples of recent articles:

·         Monahan CF, Garner MM, Kiupel M. Chromatophoromas in Reptiles. Vet Sci. 2022 Mar 4;9(3):115. doi: 10.3390/vetsci9030115. PMID: 35324843; PMCID: PMC8955407.

·         LaDouceur EE, Argue A, Garner MM. Alimentary Tract Neoplasia in Captive Bearded Dragons (Pogona spp). J Comp Pathol. 2022 Jun;194:28-33. doi: 10.1016/j.jcpa.2022.03.007. Epub 2022 Apr 25. PMID: 35577456.

·         Kleinschmidt LM, Reavill DR, Kiupel M, Hoppes SM, Strunk A, Garner MM. MAST CELL NEOPLASIA IN THE GREEN IGUANA, IGUANA IGUANA. J Zoo Wildl Med. 2023 Jan;53(4):864-869. doi: 10.1638/2022-0016. PMID: 36640091.

·         Wheelhouse JL, Mackie JT. Metastatic Anaplastic Sarcoma in a Wild Lace Monitor (Varanus varius). J Wildl Dis. 2023 Apr 1;59(2):371-375. doi: 10.7589/JWD-D-22-00106. PMID: 36989511.

·         Solanes F, Chiers K, Kik MJL, Hellebuyck T. Gross, Histologic and Immunohistochemical Characteristics of Keratoacanthomas in Lizards. Animals (Basel). 2023 Jan 24;13(3):398. doi: 10.3390/ani13030398. PMID: 36766287; PMCID: PMC9913635.

·         Monahan CF, Meyer A, Garner MM, Kiupel M. Gross, histologic, and immunohistochemical characteristics of cutaneous chromatophoromas in captive bearded dragons. J Vet Diagn Invest. 2021 Sep;33(5):932-938. doi: 10.1177/10406387211025651. Epub 2021 Jul 2. PMID: 34210217; PMCID: PMC8366258.

26- Replace ‘history’ with ‘signalment’

28- Change ‘treatments’ to ‘treatments pursued’

30-31- While these factors might be significantly associated, I’m not sure how clinically relevant they are to readers. I suspect they likely reflect lack of complete postmortem examinations or clinical staging and poor reporting in the literature, respectively, rather than significant data.

54- Add ‘comprehensive’ before ‘data’; most of the case reports that include treatment also discuss how well it worked and outcomes for that patient, so the deficit is really that larger reviews are lacking.

64- Add ‘described’ before ‘prevalence’; the true prevalence is likely much higher than what is available in the database and literature.

153-154- It would be helpful to include years in parenthesis for large number of months (ex. 300 months (25 years)), which is more tangible for most people.

Table 1- The data would be conveyed better if listed by frequency (highest to lowest) rather than alphabetically by species.

161- Table 2 is referenced, but this data is actually provided in table 1

162- Table 3 is referenced, but sex data is not reported in any of the tables. Delete in this line or add sex data if that is desired.

Table 2- 1) It would be easier for the reader if ‘benign’ or ‘malignant’ was listed after the diagnosis (ex. Change ‘benign melanophoroma to ‘melanophoroma, benign’. This would allow the reader to look at all categories of melanophoromas together. 2) Numbers under the category ‘malignancy behavior benign malignant unspecified’ need to be defined within the table or table legend OR if these are numbers affected then the table needs to be modified so that they fall under the 3 categories. It is unclear how it is currently formatted.  

177-179- How was benign versus malignant defined? Especially when so many cases lack necropsies or follow up. How was the plumed basilisk biliary adenocarcinoma classified as benign? If it is benign, it should be an adenoma instead. I’m surprised it was just 1 of unknown malignancy in the literature. If the designation was solely based on the reporting in the original reports, then this should be stated in the methods.

193- How were these cases diagnosed if ‘no treatment’ was performed? In your definition of treatments, surgery included complete and marginal excision. So, for these 159 cases only incisional biopsies were performed? If it is unclear from the report, then this should be stated instead. I imagine most of them had surgery to some extent.  

Table 4- 1) ‘Chemotherapy only’ is actually ‘surgery… followed by chemotherapy’ according to line 194-195, so this should be specified in the table, too. If this is not what is trying to be conveyed in this sentence but due to a typo then this should be corrected. 2) Survival times comparing by treatments alone isn’t very helpful because you are comparing benign and malignant processes that will skew the data. It would be more helpful to compare treatments for specific diagnosis or at least for only malignant processes to avoid data being skewed towards longer survival by presence of benign processes.

200- Similar question for ESCRA data as line 193

Table 5- Similar to table 4, the relevance for survival times based solely on treatment for various types of neoplasia does not seem clinically relevant. It would be more relevant if broken down by species or type of neoplasia or at least benign versus malignant processes. The ‘chemotherapy alone’ and ‘surgery combined with radiation therapy’ cases should be included in the table even if they are just (n=1) since they are mentioned in the paragraph.

213-214 and 222-223- Is data for survival times based on benign versus malignant diagnosis available? This could be helpful to include.

237-238- Not clinically relevant; this is just saying lack of data was significant with non-neoplastic-related outcomes.

240- Unknown status of being benign or malignant also likely isn’t all that relevant, just enough data was collected to accurately predict outcome.

Figure 1- Would be more clinically relevant if also broken down by benign versus malignant diagnoses.

257-258- Similar to comments for previously clinically irrelevant data. Lack of data variables are not helpful predictors to be included.

271-272- Similar issue as comment for line 240

279- Similar to comments for previously clinically irrelevant data. If they are going to be included, then it should be thoroughly discussed in the discussion that they are clinically irrelevant and reflect lack of consistent data being included in reports to encourage future authors to include as much data as possible. They should not be included in the abstract as great takeaways from the article.

Discussion- Also discuss small sample size biases for some reports and diagnoses.

Comments on the Quality of English Language

No comments

Author Response

Comments and Suggestions for Authors

This manuscript describes and attempts to compile a large amount of data from a diverse group of species and diagnoses. Given this diversity along with limited availability of treatment and survival data, loss to follow up, and lack of necropsies (as is typical in a large literature-based review), it is difficult to pull it all together and reach clinically significant conclusions. This report does include some relevant data, but also fairly frequently reports non-relevant data that is deemed to be statistically significant but lacks clinical significance (e.g. lack of data correlated with survival). Perhaps further clinically relevant outcomes could be reached if the data was further breakdown into diagnoses (at least benign versus malignant) or by species (or even by infraorder to increase numbers and attempt to find trends) would make it more relevant, especially for treatment and survival times. 

The following phrase ‘non- neoplasia-related outcomes of either survival or death’ is used frequently within the text and is a confusing way to convey the information. It could be re-phrased to more clearly state what is trying to be conveyed. 

  • The phrase "non-neoplasia-related outcomes of either survival or death" was changed to “non-neoplastic outcomes” with definition in abstract and results sections explaining it to mean either survival of the animal or death due to other non-neoplastic causes.

While the search parameters for the literature review component were well described, this manuscript would be improved by including more recently published papers. A quick PubMed search for ‘lizard neoplasia’ reveals multiple case reports and case series of neoplasia in lizards that have been published since August 2021. It also brought up other case reports and a few case series that were not listed in the references that should be included in this report. Below are some examples, but this is an incomplete list.

  • Thank you for bringing these publications to our attention. We have evaluated the recommended additions and made the following additions or omissions for this paper:
  • The following papers were evaluated and their information added to the statistical analysis of this paper:
    • Abou-Madi N, Kern TJ. Squamous cell carcinoma associated with a periorbital mass in a veiled chameleon (Chamaeleo calyptratus). Vet Ophthalmol. 2002 Sep;5(3):217-20
    • Schilliger L, Paillusseau C, Gandar F, De Fornel P. Iridium 192 (192-Ir) high dose rate brachytherapy in a central bearded dragon (pogona vitticeps) with rostral squamous cell carcinoma. J Zoo Wildl Med. 2020 Mar 17;51(1):241-244.
    • Williams MJ, Wong HE, Priestnall SL, Szladovits B, Stapleton N, Hedley J. Anaplastic Sarcoma and Sertoli Cell Tumor in a Central Bearded Dragon (Pogona vitticeps). J Herpetol Med Surg. 2020 Jun 11;30(2):68-73.
  • The following papers were omitted due to not having appropriate or indivisually identifying accessible information for statistical analysis:
    • Magnotti JM, Garner MM, Stahl SJ, Corbin EM, LaDouceur EEB. RETROSPECTIVE REVIEW OF HISTOLOGIC FINDINGS IN CAPTIVE GILA MONSTERS (HELODERMA SUSPECTUM) AND BEADED LIZARDS (HELODERMA HORRIDUM). J Zoo Wildl Med. 2021 Apr;52(1):166-175. doi: 10.1638/2020-0058. PMID: 33827173.
    • Mendyk RW, Newton AL, Baumer M. A retrospective study of mortality in varanid lizards (Reptilia: Squamata: Varanidae) at the Bronx Zoo: implications for husbandry and reproductive management in zoos. Zoo Biol. 2013 Mar;32(2):152-62. doi: 10.1002/zoo.21043. Epub 2012 Sep 19. PMID: 22997089.

  • The following paper was already part of the original statistical analysis for this paper:
    • Goe AM, Heard DJ, Abbott JR, de Mello Souza CH, Taylor KR, Sthay JN, Wellehan JF. Surgical management of an odontogenic tumor in a banded Gila monster (Heloderma suspectum cinctum) with a novel herpesvirus. Vet Q. 2016 Jun;36(2):109-14
  • The other recommended articles unfortunately fell outside of the scope set for this paper due to the need of establishing an appropriate end date of data collection to maintain the relatively focuses scope of this publication. We will make note of these publications for future investigations.

Examples of recent articles:

  • Monahan CF, Garner MM, Kiupel M. Chromatophoromas in Reptiles. Vet Sci. 2022 Mar 4;9(3):115. doi: 10.3390/vetsci9030115. PMID: 35324843; PMCID: PMC8955407.
  • LaDouceur EE, Argue A, Garner MM. Alimentary Tract Neoplasia in Captive Bearded Dragons (Pogona spp). J Comp Pathol. 2022 Jun;194:28-33. doi: 10.1016/j.jcpa.2022.03.007. Epub 2022 Apr 25. PMID: 35577456.
  • Kleinschmidt LM, Reavill DR, Kiupel M, Hoppes SM, Strunk A, Garner MM. MAST CELL NEOPLASIA IN THE GREEN IGUANA, IGUANA IGUANA. J Zoo Wildl Med. 2023 Jan;53(4):864-869. doi: 10.1638/2022-0016. PMID: 36640091.
  • Wheelhouse JL, Mackie JT. Metastatic Anaplastic Sarcoma in a Wild Lace Monitor (Varanus varius). J Wildl Dis. 2023 Apr 1;59(2):371-375. doi: 10.7589/JWD-D-22-00106. PMID: 36989511.
  • Solanes F, Chiers K, Kik MJL, Hellebuyck T. Gross, Histologic and Immunohistochemical Characteristics of Keratoacanthomas in Lizards. Animals (Basel). 2023 Jan 24;13(3):398. doi: 10.3390/ani13030398. PMID: 36766287; PMCID: PMC9913635.
  • Monahan CF, Meyer A, Garner MM, Kiupel M. Gross, histologic, and immunohistochemical characteristics of cutaneous chromatophoromas in captive bearded dragons. J Vet Diagn Invest. 2021 Sep;33(5):932-938. doi: 10.1177/10406387211025651. Epub 2021 Jul 2. PMID: 34210217; PMCID: PMC8366258.

26- Replace ‘history’ with ‘signalment’

  • The term was revised to “signalment”

28- Change ‘treatments’ to ‘treatments pursued’

  • The term was revised to “treatments pursued”

29-42

  • Recommended clarifications to phrasing and definition of “non-neoplastic outcomes” added.

30-31- While these factors might be significantly associated, I’m not sure how clinically relevant they are to readers. I suspect they likely reflect lack of complete postmortem examinations or clinical staging and poor reporting in the literature, respectively, rather than significant data.

  • This same association was identified in previous studies, and often aligns with a lack of treatment being pursued by the owners. The implications being that owners who do not investigate the sex of their animal often do not seek treatment, leading to a worse outcome. A paragraph has been added to the discussion to clarify this finding.

54- Add ‘comprehensive’ before ‘data’; most of the case reports that include treatment also discuss how well it worked and outcomes for that patient, so the deficit is really that larger reviews are lacking.

  • The recommended term was added

64- Add ‘described’ before ‘prevalence’; the true prevalence is likely much higher than what is available in the database and literature.

  • The recommended term was added

96-106

  • Recommended clarifications to definition malignant/benign neoplasms, as well as non-inclusion of aspirates/biopsies in treatment category added.

114

  • Recommended clarification of malignancy reporting added

122

  • Recommended clarification of non-inclusion of aspirates/biopsies in treatment category added.

153-154- It would be helpful to include years in parenthesis for large number of months (ex. 300 months (25 years)), which is more tangible for most people.

  • The recommended change was made throughout the paper

Table 1- The data would be conveyed better if listed by frequency (highest to lowest) rather than alphabetically by species.

  • The table organization theme that was determined for consistency across the paper was to record results in an alphabetical order.

161- Table 2 is referenced, but this data is actually provided in table 1

  • The recommended correction was made

162- Table 3 is referenced, but sex data is not reported in any of the tables. Delete in this line or add sex data if that is desired.

  • The reference line was removed.

Table 2- 1) It would be easier for the reader if ‘benign’ or ‘malignant’ was listed after the diagnosis (ex. Change ‘benign melanophoroma to ‘melanophoroma, benign’. This would allow the reader to look at all categories of melanophoromas together. 2) Numbers under the category ‘malignancy behavior benign malignant unspecified’ need to be defined within the table or table legend OR if these are numbers affected then the table needs to be modified so that they fall under the 3 categories. It is unclear how it is currently formatted. 

  • Thank you for this comment. We reviewed the base data and applied the recommended corrections to the indicated table. Malignancy behavior column was eliminated and replaced with represented individuals for each species affected by a neoplasm.

177-179- How was benign versus malignant defined? Especially when so many cases lack necropsies or follow up. How was the plumed basilisk biliary adenocarcinoma classified as benign? If it is benign, it should be an adenoma instead. I’m surprised it was just 1 of unknown malignancy in the literature. If the designation was solely based on the reporting in the original reports, then this should be stated in the methods.

  • Malignancy was primarily defined based on the reported behavior in the reviewed literature or submitted cases to ESCRA. Cases where malignancy was vague were determined based on generally accepted behavior of the neoplasms from reference pathologist sources. The methods section has be adjusted to clarify this point.

Table 3

  • Recommended changes to Table 2 applied. Malignancy behavior column was eliminated and replaced with represented individuals for each species affected by a neoplasm.

193- How were these cases diagnosed if ‘no treatment’ was performed? In your definition of treatments, surgery included complete and marginal excision. So, for these 159 cases only incisional biopsies were performed? If it is unclear from the report, then this should be stated instead. I imagine most of them had surgery to some extent. 

  • For how we defined treatments when initially planning this paper, we decided that surgery would include marginal/debulking excisions and higher. Sampling procedures such as fine needle aspirates and biopsies were considered diagnostic and not therapeutic, so were not included in the treatments category. A sentence has been added to the methods section to clarify this. 

Table 4- 1) ‘Chemotherapy only’ is actually ‘surgery… followed by chemotherapy’ according to line 194-195, so this should be specified in the table, too. If this is not what is trying to be conveyed in this sentence but due to a typo then this should be corrected. 2) Survival times comparing by treatments alone isn’t very helpful because you are comparing benign and malignant processes that will skew the data. It would be more helpful to compare treatments for specific diagnosis or at least for only malignant processes to avoid data being skewed towards longer survival by presence of benign processes.

  • The requested data lines were added to the table.
  • We elected to include this data as while there may be some influence from the inclusion of malignant and benign cases, the general information is still useful in a clinical sense for determining prognosis of patients when discussing survival times.

200- Similar question for ESCRA data as line 193

  • Please see above comment.

Table 5- Similar to table 4, the relevance for survival times based solely on treatment for various types of neoplasia does not seem clinically relevant. It would be more relevant if broken down by species or type of neoplasia or at least benign versus malignant processes. The ‘chemotherapy alone’ and ‘surgery combined with radiation therapy’ cases should be included in the table even if they are just (n=1) since they are mentioned in the paragraph.

  • Added requested data lines to the table.
  • See comments for Table 4.

213-214 and 222-223- Is data for survival times based on benign versus malignant diagnosis available? This could be helpful to include.

  • Unfortunately due to the retrospective nature of this data and a lack of consistency in reporting from the original sources it is not possible to perform these analyses. It is certainly relevant analysis to include in future investigations.

237-238- Not clinically relevant; this is just saying lack of data was significant with non-neoplastic-related outcomes.

  • This same association was identified in previous studies, and often aligns with a lack of treatment being pursued by the owners. The implications being that owners who do not investigate the sex of their animal often do not seek treatment, leading to a worse outcome. A paragraphs has been added to the discussion to clarify this finding.

237-257

  • Recommended clarifications to phrasing and definition of “non-neoplastic outcomes” added.

240- Unknown status of being benign or malignant also likely isn’t all that relevant, just enough data was collected to accurately predict outcome.

  • Based on the analysis for Table 2 the relevant data was updated.

Figure 1- Would be more clinically relevant if also broken down by benign versus malignant diagnoses.

  • This change was included as requested

257-258- Similar to comments for previously clinically irrelevant data. Lack of data variables are not helpful predictors to be included.

  • Please see comments for lines 237-238

Table 6

  • Results organized alphabetically for consistency

271-272- Similar issue as comment for line 240

  • Please see comment for line 240

279- Similar to comments for previously clinically irrelevant data. If they are going to be included, then it should be thoroughly discussed in the discussion that they are clinically irrelevant and reflect lack of consistent data being included in reports to encourage future authors to include as much data as possible. They should not be included in the abstract as great takeaways from the article.

  • Please see comments for line 327-328

Table 7

  • Results organized alphabetically for consistency

285-295

  • Recommended clarifications to phrasing of “non-neoplastic outcomes” added.

Table 8

  • Results organized alphabetically for consistency

Discussion- Also discuss small sample size biases for some reports and diagnoses.

335-343

  • Phrasing of results adjusted for clarity based on reviewer comments

339-352

  • Recommended clarification and framing of squamous cell carcinoma results added.

353-356

  • Phrasing of results adjusted for clarity based on reviewer comments

366-368

  • Phrasing of results adjusted for clarity based on reviewer comments

376-383

  • Discussion and framing of significant of unknown sex results added.

References

  • Additional references added:
    • Abou-Madi N, Kern TJ. Squamous cell carcinoma associated with a periorbital mass in a veiled chameleon (Chamaeleo calyptratus). Vet Ophthalmol. 2002 Sep;5(3):217-20
    • Schilliger L, Paillusseau C, Gandar F, De Fornel P. Iridium 192 (192-Ir) high dose rate brachytherapy in a central bearded dragon (pogona vitticeps) with rostral squamous cell carcinoma.
    • Newkirk KM, Brannick EM, Kusewitte DF. "Chapter 6: Neoplasia and Tumor Biology" IN Pathologic Basis of Veterinary Disease. Sixth edition. Zachary JF, ed. St. Louis, Missouri: Elsevier, 2017: 286-321.
    • Williams MJ, Wong HE, Priestnall SL, Szladovits B, Stapleton N, Hedley J. Anaplastic Sarcoma and Sertoli Cell Tumor in a Central Bearded Dragon (Pogona vitticeps). J Herpetol Med Surg. 2020 Jun 11;30(2):68-73
    • Zhender A, Graham J, Reavill D, McLaughlin A. Chapter 3 – Neoplastic diseases in avian species. Current therapy in avian medicine and surgery, 2016:107-141

Round 2

Reviewer 2 Report

Comments and Suggestions for Authors

Thank you for taking the time to make the recommended changes. The manuscript is improved by these changes. I have no further recommendations.